# Rice (*Oryza sativa* L.)–Baby Corn (*Zea mays* L.) Cropping System Response to Different Summer Green Manuring and Nutrient Management

**Vikash Kumar [1], Manoj Kumar Singh [2], Nikhil Raghuvanshi [3],* and Monalisa Sahoo [4]**

[1]   Faculty of Agricultural Science, Institute of Applied Sciences & Humanities, GLA University, Mathura 281406, India
[2]   Department of Agronomy, Banaras Hindu University, Varanasi 225001, India
[3]   Department of Agronomy, Shri Raam Swaroop Memorial University, Lucknow 2250003, India
[4]   M S Swaminathan School of Agriculture, Centurion University of Technology and Management, Paralakhemundi 761211, India
*   Correspondence: agronikhil@gmail.com

**Abstract:** The nutrient and biomass supply capacity of green manure (GM) and its complimentary and synergistic relationship with chemical nutrients is needed for a sustainable rice–baby corn cropping system in the eastern part of North India. A two-year field study was performed to assess the effect of GM and nitrogen management (NM) on yield attributes (YA), the yield of rice, and their residual (R) effect with the half-recommended dose of fertilizers on the succeeding baby corn crop. The combination of GM and N levels had significant effects on rice yield, which also influenced the succeeding baby corn crop. A higher number of tillers/m², panicle length (cm), number of grains/panicles, panicle weight (g), grain yield (Mg/ha), straw yield (Mg/ha), and biological yield (Mg/ha) were found with *Sesbania aculeata* incorporated at 45 DAS (SA), which was statistically on par with water hyacinth 5 t/ha dry weight basis (WH) during both years of the field study. Among the rates of nitrogen fertilizers, 100% RDN (50% N through FYM + 50% N through inorganic) ($RDN_{100}$) was the best treatment with a corresponding increase in all YA and yields of rice. It has been proven that integrated nitrogen management (INM), or the use of organic material along with fertilizer, is an effective way of managing nitrogen. In the present investigation, green manuring and integrated nitrogen management on rice sustainably increased the yield attributes and yield of the succeeding baby corn. These results illustrated the complementary effects of summer green manuring in conjunction with INM in maximizing the yield attributes and yields of rice, its residual effect on succeeding baby corn, and the economics of the system.

**Keywords:** integrated nitrogen management; *Sesbania aculeata*; water hyacinth; FYM; system profitability

## 1. Introduction

Farmers throughout the world are facing sustainability issues in crop production due to imbalanced and conventional practices [1]. The continuous cultivation of a cereal-based cropping system is severely affecting the crop productivity, income, and sustainability of the farm. Summer green manuring is finding prominence as a cover crop in northern plains during the summer rice–baby corn fallow period due to the limited scope of economic crop production. This practice can improve the soil organic carbon (SOC), nutrient cycling, plant growth, and soil ecosystem [2,3]. GM and NM can promote nutrition and the uptake of water, and increase the yield of crops as well as improve long-term soil fertility in sustainable crop production [4]. GM crops may be leguminous or non-leguminous and can be grown in situ or brought from outside as cuttings of trees and shrubs [5]. SA is an appropriate GM crop for rice cultivation in the Indo-Gangetic Plain. This is due to SA producing a higher biological yield in a short period, as well as tolerance to drought,

flood, and salinity [6]. Plant residue with high N concentrations, and low lignin, cellulose concentrations, and C:N ratios often result in high N mineralization. This obnoxious aquatic weed's high suitability for use as GM may be attributed to its low and narrow margin carbon–nitrogen ratio (C:N) of 1:24.3 with a lignin content of only 8–9% compared with a C:N ratio of 1:80 and lignin content of 18–20% in wheat straw [7]. The integrated application of N along with an appropriate GM may be an effective approach to enhance the efficiency of nitrogen use, soil quality, and crop yield in the GM–rice–baby corn cropping system [8]. The use of FYM alone as an alternative to inorganic fertilizer is not enough to maintain the present levels of crop productivity of high-yielding varieties [9]. Therefore, INM is required in which different organic and inorganic sources are used simultaneously, and this is the most effective technique to maintain a healthy and sustainably productive soil. Emerging evidence indicates that integrated soil fertility management involving the judicious use of combined organic and inorganic resources is a feasible approach to overcome soil fertility constraints [10,11]. The application of organic amendments such as green manuring, crop residues, and/or farmyard manure significantly increases the SOC and other essential elements in the soil for the long term. An experiment conducted on acidic soil (pH 5.4) in India reported that the application of SA along with organic manure and urea given in split doses was superior to other inorganic and organic treatments and produced a sustainable higher yield of rice [12]. A GM mixture along with organic manure could replace the 60–85 kg/ha N for succeeding crops [13]. The residual nutrient and OM could enhance the growth and yield of succeeding crops. INM with inorganic and organic manure along with green manure is a sustainable practice that would result in 25–50% economic savings on the cost of fertilizers under a rice-based system [14]. This study aimed to investigate (1) the sustainable and economical options of GM and (2) the effect of SGM and NM on yield attributes and the yield of rice and succeeding crops, and (3) to analyze the economics and system productivity and profitability.

## 2. Material and Method

### 2.1. Field Experiments

The field experiments were carried out from April 2018 to February 2020 at the Agriculture Research Station, Banaras Hindu University, Varanasi, India, on sandy loam soil. The experiment was laid out in split-plot design (SPD) assigning four GM treatments in the main plot, viz., fallow (F), water hyacinth 5 t/ha (dry weight basis) (WH), *Sesbania aculeata* incorporated at 45 DAS (SA), Sudan grass incorporated at 45 DAS (SU), and three nitrogen management in subplot, viz., 60% RDN (50% N through FYM + 50% N through inorganic) (RDN$_{60}$), 80% RDN (50% N through FYM + 50% N through inorganic) (RDN$_{80}$), 100% RDN (50% N through FYM + 50% N through inorganic) (RDN$_{100}$). Before the start of the experiment, the rice–wheat cropping system was cultivated each cropping year. Green manuring crops, viz., SA and SU were sown in situ in the first week of May and harvested 45 days after sowing. No fertilizers were applied for GM cultivation. The WA was collected from a pond at the time of harvesting other GM crops. The nursery of hybrid rice (*Oryza sativa* L.) was sown in the third week of June in a separate field. Fertilizers for the nursery were applied at the rate of 10 t/ha FYM, 20 kg/ha nitrogen, 20 kg/ha phosphorous, and 10 kg/ha zinc sulphate at the sowing. Two weeks later 20 kg/ha nitrogen through urea was applied as a top dressing after irrigating the nursery. Healthy plants at 20–25 days in the nursery were transplanted in the study field in the last week of July at the distance of 20 cm row to row and 15 cm plant to plant to plant. Field was ploughed with soil turning plough and furrow made by ridge furrow maker, then all the GM treatments were placed in the furrow and then covered by soil for decomposition. Nutrients and biomass were added before transplanting was varied and calculated before incorporating the green manuring, given below in Table 1. A uniform dose at the rate of 75–60–5 of phosphorous, potash, and zinc, respectively, was applied through DAP, MOP, and zinc sulphate for rice crop, and half dose of RDF (60–30–20) of NPK was applied in baby corn. Twenty-five percent chemical nitrogen and 50 percent through FYM, and a full dose of phosphorous, potassium, and zinc

were applied at the time of sowing, and the remaining half dose of nitrogen was applied at the booting stage in hybrid rice. In the case of baby corn cultivation, a half dose of nitrogen and a full dose of phosphorous and potash were applied at the time of sowing, and the remaining half dose of nitrogen was applied 30 days after sowing. Four irrigations in rice and five irrigations in baby corn were applied with one pre-sowing irrigation. The hybrid rice was harvested in October and YA, grain, and straw yield were measured at the time of harvesting.

**Table 1.** Nutrient status of different organic materials applied in experimental field.

| Particular | *Sesbania aculeata* (45 DAS) | | Water Hyacinth (5 t/ha DWB) | | Sudan Grass (45 DAS) | | FYM | |
|---|---|---|---|---|---|---|---|---|
| Nitrogen content (%) | 2.3 | 2.5 | 2.4 | 2.3 | 0.36 | 0.38 | 0.48 | 0.50 |
| Phosphorous content (%) | 0.70 | 0.76 | 0.56 | 0.72 | 0.26 | 0.27 | 0.25 | 0.26 |
| Potassium content (%) | 1.21 | 1.32 | 0.32 | 0.38 | 0.84 | 0.92 | 0.52 | 0.48 |
| Dry Biomass (kg/ha) | 4000 | 4240 | 5000 | 5000 | 7222.2 | 6052.6 | - | - |
| Nitrogen amount (kg/ha) | 92 | 106 | 120 | 115 | 26 | 23 | 45, 60, and 75 | 45, 60, and 75 |
| Moisture content | 86.7 | 88.9 | 93.2 | 92.2 | 83.7 | 85.2 | - | - |

*2.2. Sampling and Measurement*

From the treatment plot, 10 samples were collected randomly and the values of individuals for YA in hybrid rice were averaged. Rice equivalent yield (REY, q/ha), system productivity (SP, kg/ha/day), system profitability (SPE, USD/ha/day), and land-use efficiency (LUE, %) were calculated using given formula. The linear relationship of hybrid rice yield and baby corn with their yield attributes was calculated with standard formula and illustrated in the figures. The economics of the cropping system was calculated by prevailing prices of inputs used and the output realized. The costs of cultivation of separate crops as well as GM were individually calculated and the total cost of cultivation for the crop system was calculated. The yields of the hybrid rice and baby corn crop system were converted into gross returns in rupees. Furthermore, by the below-given formula, net returns and B:C (benefit–cost) ratio were also calculated.

$$Net\ returns\ (Mg/ha) = Gross\ returns - cost\ of\ cultivation \tag{1}$$

$$B:C\ ratio = \frac{Net\ returns\ of\ system\ (Mg/ha)}{Cost\ of\ cultivation\ of\ system\ (Mg/ha)} \tag{2}$$

$$Rice\ equivalent\ yield = \frac{Yield\ of\ baby\ corn\ \left(\frac{Mg}{ha}\right) \times Price\ of\ baby\ corn\ (Mg/ha)}{Price\ of\ rice\ (Mg/ha)} \tag{3}$$

*2.3. Climatic Condition of the Experimental Site*

The experimental site comes under the Northern Gangetic Alluvial Plain of India (83°03′0″ E longitude; 25°18′0″ N latitude and an altitude of 128.9 m above sea level). The experimental site was situated in the alluvial soil zone of eastern Uttar Pradesh. The experimental site is situated in a semi-arid to subhumid environment with a moisture deficit index between 20 to 40%. The temperature ranged from 27.3 °C to 41 °C in summer and from 21.9 °C to 3.9 °C in winter in 2018, whereas it ranged from 27.1 °C to 43.8 °C in summer 2019 and from 34.2 °C to 7 °C in winter 2019. The average annual rainfall is about 1100 mm of which more than 70% occurs in the monsoon season from July to September. The climate of the experimental area is divided into three seasons, viz., monsoon or rainy season (June to September), winter season (October to February), and summer season

(March to May). The monthly average temperature and rainfall of the crop growing period are illustrated in the Figure 1.

**Figure 1.** Weather condition of experimental field.

### 2.4. Statistical Analysis

The data pertaining to yield attributes and economics were analyzed in split-plot design using R (R × 64 4.0.2) and R studio (Version 1.3.1093 R Studio, PBC; Vienna, Austria) [15] packages bioresearch, Agricolae, and least significant difference (LSD) values were tested at 5% level of significance ($p = 0.05$). Correlation and linear regression analysis were performed by methodology given in by using R programming [16]. Linear regression analyses ($p < 0.05$) were performed to determine the relationships among the YA and yield of rice, the YA, and yield of baby corn, economics, and system productivity.

### 3. Results

### 3.1. Effect of GM and NM on Yield Attributes and Yield of Hybrid Rice

The mean data from the two years of the experiment were used and variability between the years and treatments was not found in the present study for YA and Y of HR. During hybrid rice cultivation, the yield was increased from 5.21 Mg/ha to 7.09 Mg/ha by green manuring and from 5.24 Mg/ha to 6.63 Mg/ha due to NM. All yield attributes and the yields of hybrid rice were significantly affected due to SGM and NM, except for unfilled spikelets per spike, test weight, and harvest index. Data about the yield attributes and yields are presented in Table 2 indicating that all the treatments proved their significant superiority in higher fertility conditions. It is evident that the higher yield attributes and yield found in the second year in comparison to the first year may be due to meteorological conditions and higher soil fertility. During the study, the maximum YA and Y values were obtained from SA, which was statistically comparable with WH, whereas, the minimum YA and Y were observed with F, which was followed by SG. Legume GM with SA increased the sustainable yield component and yield of crops in comparison to non-legumes or fallow and reduced the cost of cultivation of the system. Among the NM treatments, the superior YA and Y were observed under higher fertility levels ($RDN_{100}$) followed by $RDN_{80}$. The yield of hybrid rice increased under a higher fertility level. The research finding reported that the application of nitrogen with the combination of chemical and organic treatments reduced the chemical nitrogen application.

**Table 2.** Effect of summer green manuring and nitrogen management on yield attributes of unpuddled transplanting of hybrid rice.

| Treatments | No. of Tillers | Panicle Length (cm) | Unfilled Spikelet/Spike | Number of Grain/Spikes | Test Weight (g/1000 Seed) | Panicle Weight (g) | Grain Yield (Mg/ha) | Straw Yield (Mg/ha) | HI |
|---|---|---|---|---|---|---|---|---|---|
| **Green manuring** | | | | | | | | | |
| F | 220.80± 13.2 c | 20.26 ± 0.80 b | 38.87 ± 1.42 b | 64.58 ± 4.02 b | 24.02 ± 0.015 ab | 5.03 ± 0.25 b | 5.21 ± 0.38 b | 6.65 ± 0.49 b | 43.82 ± 0.56 a |
| WA | 306.68 ± 9.23 b | 25.68 ± 1.08 a | 43.59 ± 0.89 ab | 91.75 ± 5.02 a | 24.16 ± 0.19 a | 6.06 ± 0.22 a | 6.53 ± 0.26 a | 8.42 ± 0.35 a | 43.69 ± 0.20 a |
| SA | 339.02 ± 8.35 a | 26.30 ± 0.63 a | 46.22 ± 2.87 a | 97.52 ± 2.56 a | 24.29 ± 0.18 a | 6.48 ± 0.23 a | 7.09 ± 0.25 a | 9.08 ± 0.25 a | 43.85 ± 0.24 a |
| SU | 205.81 ± 6.26 c | 19.85 ±1.05 b | 44.71 ± 183 a | 61.70 ± 3.66 b | 23.76 ± 0.11 b | 5.10 ± 0.15 b | 5.33 ± 0.21 b | 6.88 ± 0.29 b | 43.64 ± 0.23 a |
| *p*-value | <0.001 *** | 0.003 ** | 0.0559 | <0.001 *** | 0.097 ns | 0.002 ** | <0.001 ** | <0.001 ** | 0.935 ns |
| CD | 30.35 | 3.09 | ns | 8.98 | ns | 0.606 | 0.736 | 0.926 | ns |
| CV | 9.82 | 11.63 | 10.37 | 9.87 | 1.39 | 9.26 | 10.55 | 10.34 | 1.89 |
| **Nitrogen management** | | | | | | | | | |
| RDN$_{60}$ | 226.48 ± 11.70 c | 20.66 ± 1.01 c | 41.20 ± 1.55 b | 66.29 ± 2.91 c | 23.90 ± 0.21 b | 5.12 ± 0.17 b | 5.24 ± 0.18 b | 6.73 ± 0.24 c | 43.78 ± 0.23 a |
| RDN$_{80}$ | 268.20 ± 8.27 b | 23.51 ± 0.69 b | 43.41 ± 1.80 ab | 80.80 ± 4.39 b | 24.11 ± 0.017 ab | 5.80 ± 0.22 a | 6.25 ± 0.36 a | 8.01 ± 0.41 b | 43.80 ± 0.31 a |
| RDN$_{100}$ | 309.55 ± 7.80 a | 24.91± 0.98 a | 45.44 ± 1.91 a | 89.57 ± 3.50 a | 24.16 ± 0.10 a | 6.08 ± 0.25 a | 6.63 ± 0.36 a | 8.53 ± 0.40 a | 43.67 ± 0.38 a |
| *p*-value | <0.001 *** | <0.001 *** | 0.004 ** | <0.001 *** | 0.066 ns | <0.001 *** | <0.001 *** | <0.001 *** | 0.772 ns |
| CD | 11.70 | 0.99 | 2.28 | 4.60 | ns | 0.296 | 0.454 | 0.520 | ns |
| CV | 5.04 | 4.97 | 6.08 | 6.74 | 0.975 | 6.08 | 8.67 | 7.74 | 1.07 |
| Interaction | <0.001 *** | ns | 0.003 ** | 0.006 ** | ns | 0.038 * | 0.022 * | 0.010 * | ns |
| Year | ns | ns | ns | ns | ns | ns | ns | ns | ns |

Note: F, fallow; WA, water hyacinth 5 t/ha (dry weight basis); SA, *Sesbania aculeata* incorporated at 45 DAS; SU, Sudan grass incorporated at 45 DAS, and three nitrogen management: (RDN$_{60}$), 60% RDN (50% N through FYM + 50% N through inorganic); (RDN$_{80}$), 80% RDN (50% N through FYM + 50% N through inorganic); (RDN$_{100}$); 100% RDN (50% N through FYM + 50% N through inorganic) (RDN$_{100}$). Data followed by different letters are significantly different with confidence level set at 0.05. ns, not significant; * $p < 0.05$; ** $p < 0.01$ and *** $p < 0.001$.

### 3.2. Residual Effect of GM and NM with 50% RDN on Yield Attributes (YA) and Yield (Y) of Baby Corn

The SGM and FYM slowly release the nutrients into the soil due to slow mineralization; therefore, organic manuring did not enhance the yield of the main crop but increased the yield of succeeding crops. Under our study, the baby corn yield (Mg/ha) was increased with residual GM and FYM and the available soil nutrients. Data related to the residual effect of green manuring and nitrogen management on yield attributes and the yield of baby cobs are presented in Tables 3 and 4. A higher baby corn yield was counted with residual nutrients and biomass added by SA with 50% RDN, which was statistically similar to residual WH with 50% RDN under the field investigation, while the minimum baby corn YA and Y was obtained with F with 50% RDN, which was followed by residual SG with 50% RDN. The analysis of data shows that different residual N caused the variation in the YA and Y of baby corn. The higher YA and Y were recorded with residual $RDN_{100}$ with 50% RDN, which was statistically on par with residual $RDN_{80}$ with 50% RDN.

**Table 3.** Effect of summer green manuring and nitrogen management on yield attributes of succeeding baby corn.

| Treatments | No. of Cobs/Plant | Bareness (%) | Cob Length (cm) | Corn Length (cm) | Cob Girth (cm) | Corn Girth (cm) | Cob Weight (g) | Corn Weight (g) |
|---|---|---|---|---|---|---|---|---|
| | | | **Green manuring** | | | | | |
| F | 1.83 ± 0.18 b | 14.62 ± 1.36 a | 23.00 ± 2.35 b | 6.30 ± 0.44 c | 7.79 ± 0.48 c | 3.80 ± 0.14 c | 27.76 ± 0.99 b | 6.91 ± 0.30 b |
| WA | 2.44 ± 0.20 a | 10.98 ± 0.96 b | 33.39 ± 2.06 a | 7.90 ± 0.69 ab | 9.50 ± 0.50 ab | 4.59 ± 0.28 ab | 36.18 ± 1.54 a | 8.82 ± 0.39 a |
| SA | 2.54 ± 0.24 a | 8.10 ± 1.05 c | 35.61 ± 0.97 a | 8.34 ± 0.47 a | 10.59 ± 0.43 a | 4.90 ± 0.22 a | 38.00 ± 0.74 a | 9.49 ± 0.0.49 a |
| SU | 1.90 ± 0.16 b | 14.52 ± 2.42 a | 26.33 ± 1.95 b | 6.51 ± 0.29 bc | 8.73 ± 0.39 bc | 4.01 ± 0.20 bc | 29.57 ± 0.77 b | 7.10 ± 0.37 b |
| *p*-value | 0.003 ** | 0.003 ** | 0.005 ** | 0.034 * | 0.024 * | 0.011 * | 0.000 *** | 0.002 ** |
| CD | 0.27 | 2.77 | 5.79 | 1.46 | 1.58 | 0.581 | 3.38 | 1.08 |
| CV | 10.75 | 19.89 | 16.96 | 17.44 | 14.99 | 11.64 | 8.92 | 11.67 |
| | | | **Nitrogen management** | | | | | |
| $RDN_{60}$ | 1.98 ± 0.20 b | 14.23 ± 1.93 a | 26.70 ± 2.08 c | 6.66 ± 0.48 b | 8.29 ± 0.48 c | 4.00 ± 0.24 b | 27.96 ± 0.67 | 6.78 ± 0.40 c |
| $RDN_{80}$ | 2.18 ± 0.19 a | 12.08 ± 1.38 ab | 29.78 ± 1.74 b | 7.23 ± 0.49 b | 9.18 ± 0.49 b | 4.33 ± 0.20 ab | 33.83 ± 1.59 | 8.39 ± 0.35 b |
| $RDN_{100}$ | 2.36 ± 0.19 a | 9.85 ± 1.04 b | 32.29 ± 1.70 a | 7.90 ± 0.44 a | 9.98 ± 0.37 a | 4.61 ± 0.20 a | 36.84 ± 0.78 | 9.07 ± 0.42 a |
| *p*-value | <0.001 ** | 0.009 ** | <0.001 *** | 0.002 ** | <0.001 *** | 0.003 ** | <0.001 *** | <0.001 *** |
| CD | 0.18 | 2.61 | 2.25 | 0.615 | 0.558 | 0.29 | 1.29 | 0.49 |
| CV | 9.34 | 25.06 | 8.79 | 9.79 | 7.04 | 7.91 | 4.53 | 7.08 |
| Interaction | 0.001 ** | ns | ns | ns | ns | ns | 0.004 ** | ns |
| Year | ns | ns | ns | ns | ns | ns | ns | ns |

Note: F, fallow; WA, water hyacinth 5 t/ha (dry weight basis); SA, *Sesbania aculeata* incorporated at 45 DAS; SU, Sudan grass incorporated at 45 DAS, and three nitrogen management: ($RDN_{60}$), 60% RDN (50% N through FYM + 50% N through inorganic); ($RDN_{80}$), 80% RDN (50% N through FYM + 50% N through inorganic); ($RDN_{100}$); 100% RDN (50% N through FYM + 50% N through inorganic) ($RDN_{100}$) CD, Critical Difference, CV, coefficient of variation. Data followed by different letters are significantly different with confidence level set at 0.05. ns, not significant; * $p < 0.05$; ** $p < 0.01$ and *** $p < 0.001$.

### 3.3. Effect of GM and NM on the Economics of the CS

Data related to the effect of green manuring and nitrogen management on the economics of the rice–baby cob cropping system are given in Table 5. The highest total cost of cultivation was found under SG, which was followed by SA. GM with SA recorded higher gross returns, net returns, and B:C ratios, which was followed by WH and lower returns were recorded under F. Under nitrogen management treatment, the higher cost of cultivation, higher gross returns, net returns, and B:C ratio of the rice–baby corn cropping system were observed under $RDN_{100}$, which was followed by $RDN_{80}$. In the case of the interaction effect of the system, maximum gross returns, net returns, and B:C ratios were observed under the SA + $RDN_{100}$ combination, which was statistically on par with WH + $RDN_{100}$ followed by SA + $RDN_{80}$, which was significantly superior to other treatment combinations under the field study.

**Table 4.** Effect of summer green manuring and nitrogen management on yield attributes and system efficiencies of succeeding baby corn.

| Treatments | Cob Yield (Mg/ha) | Corn Yield (Mg/ha) | Fodder Yield (Mg/ha) | HI | REY (Mg/ha) | SP (kg/ha/day) | SPE ($/ha/day) | LUE (%) |
|---|---|---|---|---|---|---|---|---|
| | | | | Green manuring | | | | |
| F | 3.17 ± 2.35 b | 0.890 ± 0.096 c | 18.21 ± 1.25 b | 4.68 ± 0.31 a | 7.007 ± 0.75 c | 83.03 ± 8.10 b | 12.458 ± 1.59 b | 0.594 ± 0.005 c |
| WA | 41.56 ± 2.22 a | 1.224 ± 0.090 ab | 23.57 ± 0.58 a | 4.94 ± 0.20 a | 9.624 ± 0.70 ab | 112.14 ± 7.78 a | 17.517 ± 1.32 a | 0.592 ± 0.003 c |
| SA | 42.98 ± 2.20 a | 1.297 ± 0.028 a | 24.54 ± 0.54 a | 5.03 ± 0.05 a | 10.189 ± 0.22 a | 116.02 ± 2.36 a | 16.120 ± 0.46 a | 0.712 ± 0.004 b |
| SU | 33.72 ± 2.34 b | 1.009 ± 0.051 bc | 19.96 ± 0.88 b | 4.85 ± 0.17 a | 7.867 ± 0.40 bc | 89.39 ± 4.41 b | 10.812 ± 0.70 b | 0.725 ± 0.002 a |
| *p*-value | 0.002 ** | 0.014 * | 0.002 ** | ns | 0.0132 * | 0.013 * | 0.008 ** | <0.001 *** |
| CD | 4.79 | 0.225 | 2.42 | ns | 1.763 | 19.21 | 3.473 | 0.006 |
| CV | 11.08 | 17.74 | 9.74 | 11.49 | 17.62 | 16.62 | 21.013 | 0.780 |
| | | | | Nitrogen management | | | | |
| $RDN_{60}$ | 32.92 ± 1.87 b | 0.967 ± 0.045 b | 19.05 ± 0.42 c | 4.87 ± 0.17 a | 7.605 ± 0.35 b | 87.58 ± 3.81 b | 11.809 ± 0.56 c | 0.659 ± 0.004 a |
| $RDN_{80}$ | 38.84 ± 2.28 a | 1.332 ± 0.080 a | 22.21 ± 0.92 b | 4.84 ± 0.19 a | 8.900 ± 0.63 a | 102.93 ± 6.80 a | 14.891 ± 1.25 b | 0.655 ± 0.003 a |
| $RDN_{100}$ | 40.77 ± 2.83 a | 1.209 ± 0.074 a | 23.44 ± 1.12 a | 4.92 ± 0.19 a | 9.510 ± 0.58 a | 109.93 ± 6.37 a | 16.286 ± 1.24 a | 0.654 ± 0.003 a |
| *p*-value | <0.001 *** | <0.001 *** | <0.001 *** | ns | <0.001 *** | <0.001 *** | <0.001 *** | ns |
| CD | 2.24 | 0.085 | 1.14 | ns | 0.667 | 7.31 | 1.375 | ns |
| CV | 6.91 | 8.95 | 6.13 | 5.41 | 8.894 | 8.43 | 11.093 | 1.129 |
| Interaction | 0.009 ** | ns | 0.005 ** | ns | ns | ns | 0.019 * | ns |
| Year | | ns | ns | ns | ns | ns | 0.012 * | 0.013 * |

Note: F, fallow; WA, water hyacinth 5 t/ha (dry weight basis); SA, *Sesbania aculeata* incorporated at 45 DAS; SU, Sudan grass incorporated at 45 DAS, and three nitrogen management: ($RDN_{60}$), 60% RDN (50% N through FYM + 50% N through inorganic); ($RDN_{80}$), 80% RDN (50% N through FYM + 50% N through inorganic); ($RDN_{100}$); 100% RDN (50% N through FYM + 50% N through inorganic) ($RDN_{100}$) CD, Critical Difference, CV, coefficient of variation. Data followed by different letters are significantly different with confidence level set at 0.05. ns, not significant; * $p < 0.05$; ** $p < 0.01$ and *** $p < 0.001$.

**Table 5.** Effect of summer green manuring and nitrogen management on economics of unpuddled transplanting of hybrid rice and baby corn cropping system.

| Treatments | Cost of Cultivation (USD/ha) | Gross Returns (USD/ha) | Net Returns (USD/ha) | B:C Ratio |
|---|---|---|---|---|
| | | Green manuring | | |
| F | 1428.39 d | 4130.04 ± 350.4 b | 2701.65 ± 350.4 b | 1.890 ± 0.24 b |
| WA | 1563.52 c | 5434.80 ± 268.7 a | 3871.27 ± 268.7 a | 2.471 ± 0.17 a |
| SA | 1591.90 b | 5781.30 ± 133.7 a | 4189.40 ± 133.7 a | 2.626 ± 0.08 a |
| SU | 1600.04 a | 4464.51 ± 190.9 b | 2864.49 ± 190.3 b | 1.788 ± 0.11 b |
| *p*-value | <0.001 *** | <0.001 ** | 0.006 ** | 0.016 * |
| CD | - | 750.29 | 750.29 | 0.509 |
| CV | - | 13.133 | 19.09 | 20.12 |
| | | Nitrogen management | | |
| $RDN_{60}$ | 1522.51 c | 4337.52 ± 131.4 c | 2815.00 ± 131.4 b | 1.849 ± 0.08 b |
| $RDN_{80}$ | 1545.97 b | 5098.43 ± 294.1 b | 3552.46 ± 294.1 a | 2.288 ± 0.19 a |
| $RDN_{100}$ | 1569.39 a | 5422.04 ± 282.2 a | 3852.65 ± 282.2 a | 2.444 ± 0.1 a |
| *p*-value | <0.001 *** | <0.001 *** | <0.001 *** | <0.001 *** |
| CD | - | 319.29 | 638.38 | 0.202 |
| CV | - | 7.447 | 10.82 | 10.65 |
| Interaction | 0.000 *** | 0.018 * | 0.018 * | 0.021 * |
| Year | ns | 0.012 * | 0.012 * | 0.012 * |

Note: F, fallow; WA, water hyacinth 5 t/ha (dry weight basis); SA, *Sesbania aculeata* incorporated at 45 DAS; SU, Sudan grass incorporated at 45 DAS, and three nitrogen management: ($RDN_{60}$), 60% RDN (50% N through FYM + 50% N through inorganic); ($RDN_{80}$), 80% RDN (50% N through FYM + 50% N through inorganic); ($RDN_{100}$); 100% RDN (50% N through FYM + 50% N through inorganic) ($RDN_{100}$) CD, Critical Difference, CV, coefficient of variation. Data followed by different letters are significantly different with confidence level set at 0.05. ns, not significant; * $p < 0.05$; ** $p < 0.01$ and *** $p < 0.001$.

### 3.4. System Observation

GM not only increased the yield of the main crop, but it also influenced the productivity and profitability of the system. The data from the system observation are given in Table 4. In our studies, GM with SA recorded higher REY, SPr, and SPf values, which were statistically on par with WH. However, the lowest values were obtained by F, which was statistically on par with SG under the examination, whereas, the REY, SPr, and SPf values were significantly influenced by higher fertility levels, and $RDN_{100}$ had significantly higher values, which were statistically comparable with $RDN_{80}$. The interaction effects of GM and NM for REY, SPr, and SPf were found to be significantly affected, and higher values were observed under SA along with the $RDN_{100}$ combination. It was statistically on par with WH along with $RDN_{80}$ and SA + $RDN_{80}$, which was significantly superior to other treatment combinations during both years of the field study.

### 3.5. Relationship of Yield Component and Yield of Rice–Baby Corn System

Linear regression and correlation analysis results between the yield and yield attributes and system profitability to find out the relationship between the rice and baby corn system are given in Figures 2 and 3. The relationship between system profitability with different rice and baby corn YA and Y was studied using regression models. Results showed that the system profitability had a strongly significant (>0.05) linear regression relationship with the numbers of tillers/$m^2$, panicle length, panicle weight, grain yield, and straw yield. The small *p*-value (<0.001) indicates strong evidence against the null hypothesis among the different treatments, so the null hypothesis was rejected; however, a non-significant relationship was found for the unfilled spikelet/spike. In addition to these, as shown on Table 6, it was observed that the yield of rice had a significantly (<0.05) positive correlation with the panicle weight (g), number of grain/panicles, panicle length, and number of tillers, but the test weight and unfilled spikelet were significantly (>0.05) correlated with the grain yield of rice. The direct effect of INM was observed with the succeeding baby corn crop. A strong significant linear relationship (*p* = 0.001) (Figure 3) was observed between system profitability and corn weight, corn length, corn yield, and fodder yield. The system profitability was increased with different yield components of baby corn, indicating that the SPf directly correlates with the yield components and yield of baby corn. A strong significant relationship was found under different YA and Y of corn. Results in Table 7 showed that corn yield (Mg/ha) had a significant positive (*p* = 0.001) correlation with the weight of corn, weight of cob, corn girth, cob girth, length of corn, length of cob, and the number of cobs/plant. However, bareness showed a negative correlation with all the yield components as well as corn yield.

**Table 6.** Correlations between yield and yield attributes of hybrid rice.

| Components | Straw Yield (Mg/ha) | Grain Yield (Mg/ha) | Panicle Weight (g) | Test Weight (g) | Number of Grains/Panicles | Unfilled Spikelet/Plant | Panicle Length (cm) |
|---|---|---|---|---|---|---|---|
| Grain yield (Mg/ha) | 0.994 *** | | | | | | |
| Panicle weight (g) | 0.926 *** | 0.926 *** | | | | | |
| Test weight (g) | 0.414 * | 0.415 * | 0.380 * | | | | |
| Number of grains/panicles | 0.925 *** | 0.917 *** | 0.873 *** | 0.541 ** | | | |
| Unfilled spikelet/Plant | 0.407 ** | 0.398 ** | 0.420 * | 0.266 ns | 0.417 * | | |
| Panicle length (cm) | 0.838 *** | 0.817 *** | 0.837 *** | 0.462 ** | 0.916 *** | 0.382 * | |
| Number of tillers/$m^2$ | 0.901 *** | 0.882 *** | 0.860 *** | 0.601 *** | 0.966 *** | 0.395 * | 0.902 * |

**Note:** ns, not significant; * *p* < 0.05; ** *p* < 0.01 and *** *p* < 0.001.



**Table 7.** Correlations between corn yield and yield attributes of baby corn.

| Components | Yield of Corn (Mg/ha) | Weight of Corn | Weight of Cob | Corn Girth | Cob Girth | Length of Corn | Length of Cob | Bareness (%) |
|---|---|---|---|---|---|---|---|---|
| Weight of Corn (g) | 0.800 *** | | | | | | | |
| Weight of Cob (g) | 0.829 *** | 0.923 *** | | | | | | |
| Corn Girth (cm) | 0.713 *** | 0.721 *** | 0.743 *** | | | | | |
| Cob Girth (cm) | 0.726 *** | 0.710 *** | 0.735 *** | 0.919 *** | | | | |
| Length of Corn (cm) | 0.670 *** | 0.626 *** | 0.597 *** | 0.718 *** | 0.677 *** | | | |
| Length of Cob (cm) | 0.675 *** | 0.772 *** | 0.754 *** | 0.811 *** | 0.844 *** | 0.702 *** | | |
| Bareness (%) | −0.575 *** | −0.686 *** | −0.749 *** | −0.616 *** | −0.685 *** | −0.486 * | −0.730 *** | |
| Number of cobs/plants | 0.703 *** | 0.648 *** | 0.739 *** | 0.769 *** | 0.736 *** | 0.526 * | 0.650 *** | −0.617 *** |

* $p < 0.05$ and *** $p < 0.001$.

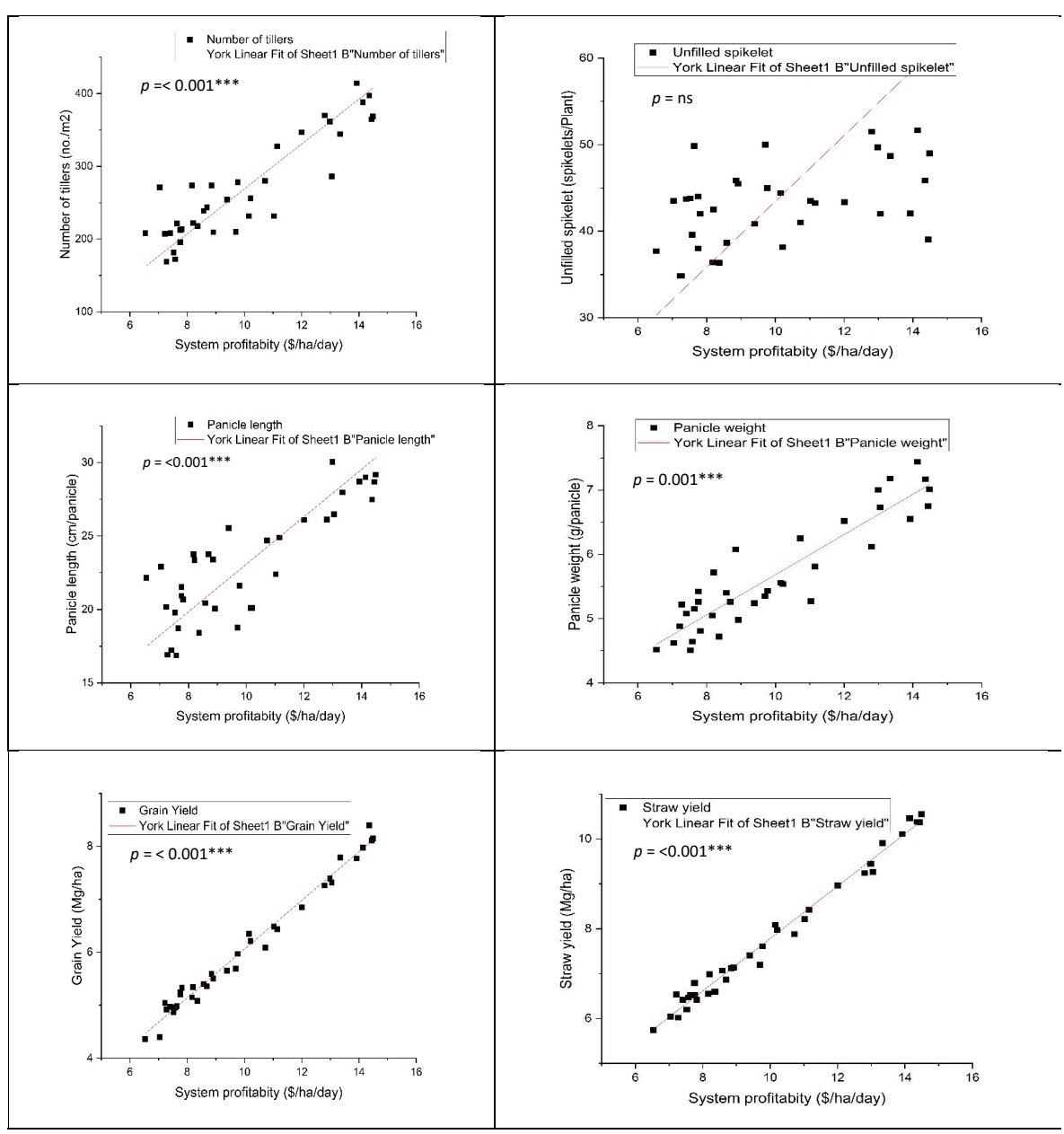

**Figure 2.** Relationship between system profitability and yield attributes and yield of unpuddled transplanted rice. Note: *** represent significant difference at the <0.001 levels.

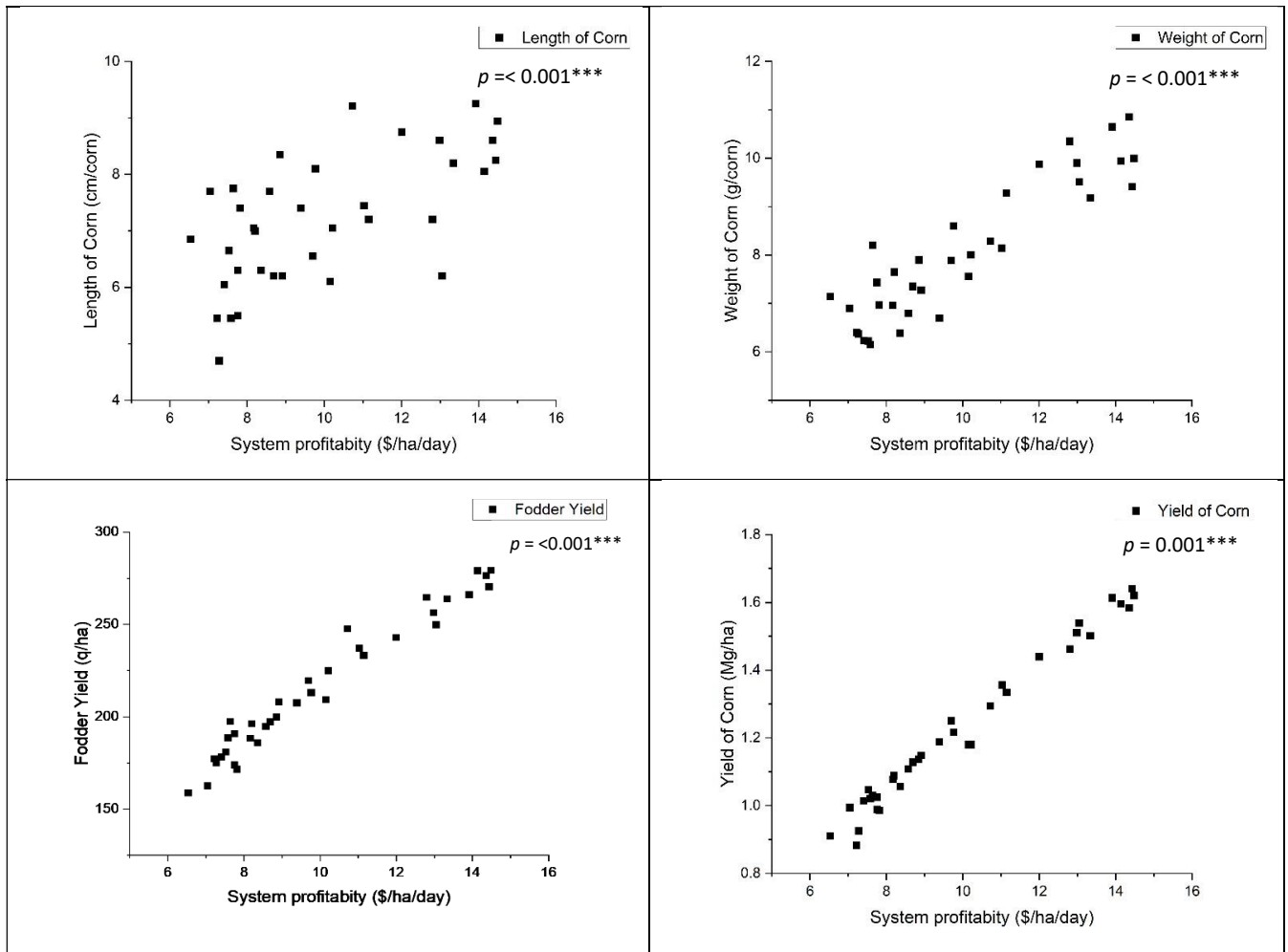

**Figure 3.** Relationship between system profitability and yield attributes and yield of baby corn. Note: *** represent significant difference at the <0.001 levels.

## 4. Discussion

It is evident from the research findings that GM and NM significantly increased the YA, Y, and economics of hybrid rice and succeeding baby corn. GM is one of the most important practices in India had been an indispensable practice since ancient times. In the present study, the application of GM with the rice–baby corn cropping system with SA had a higher yield for both crops. This finding was also reported by [17] who reported that the SA incorporated into the soil at 46 days after sowing (DAS) contributes 2.26 Mg/ha dry biomass and 76 kg/ha N to the next rice crop, and the application significantly increased grain yield by 159% and total dry biomass yield by 140–160% more than no GM. The same finding was suggested by [18,19]. In China as reported by [20], the 4 year mean yields of rice were increased by 18.6%, 8.5%, 12.3%, 14.6%, and 24.1% for LGM Chinese milky vetch (CMv), respectively, when applying early-season rice straw (Ms), applying early-season and late-season rice straw (DsMs), and applying CMv and Ms, CMv, and DsMs treatments with respect to unamended organic amendment (CK). A higher dose of nitrogen did not increase the yield as the rice grain and straw yield [21–24] were found to be similar for both the N levels of 100% RND and 125% RND, which were significantly higher than the N level of 75% RND. Rice yield and nitrogen use efficiency both exhibited a tendency for growing then declining with higher nitrogen fertilizer application, while nitrogen loss increased steadily with increasing nitrogen fertilizer input [25]. The combination of GM with SA and inorganic fertilizer helped to reduce energy input (24%), and enhance rice

yield and net returns (8%). Thus, the application of SA as a GM with the CS saves the fertilizers' consumption and maintains the long-term soil fertility [6]. Chinese milk vetch incorporation within the soil as a GM reduced chemical fertilizer input by 20–40%. This is possible because of the nitrogen fixation performed by the root nodules of Chinese milk vetch, which may have the potential to increase the productivity of rice, which, in turn, improves the sustainability yield index [26]. Integration of LGM into the existing CS significantly increased the yield of maize, suppressed the weed population, and increased N accumulation compared to fallow [27]. The long-term effect of the rice–rice–LGM CS could significantly enhance the nitrogen and phosphorus balance and their association with soil nitrogen and phosphorus content, respectively [28]. The study highlights that GM is an important practice for the agro-ecological environment and soil fertility in a double rice CS in red paddy soil. The incorporation of legume GM in the soil increased the OM in the soil and made nutrients readily available to the crops by improving the soil eco-service. Growing SA as a GM in loam sandy soil, and the incorporation of GM before sowing rice and maize, increased the water-stable aggregates by 62%, the size between 0.1 and 0.5 mm, reduced the soil bulk density, and increased the infiltration rate [29]. GM increased the OM in the soil, which helps to retain the water in the soil that helps to increase the water-holding capacity of the soil. Clover catch crop as a GM had the potential to replace the chemical fertilizer (CF) and manure without a loss in yield and lower N and K leaching in the CS [30]. WH compost increased the morphological, physiological character, and biological yield of black gram. WH is the most hazardous weed of water bodies, but it has the potential to be used as a GM, compost, or in other ways due to its lower C:N ratio and higher nutrient content [31]. WH manure is an economical and feasible technology for farmers, and 100% WH manure significantly enhances the soil productivity of crop plants [32]. We also observed in our research, the incorporation of WH as green manure improved the soil ecosystem, and during the first year of GM, about 40–45% of organic N in organic manures may be expected to become available for plant nutrition [6] and, hence, may be an alternative option for farmers to other GM crops in areas where WH is widely available and is a more problematic weed. WH is a nutritious weed with a lower C:N ratio and has the opportunity to enhance the production of rice and succeeding baby corn. Manure application alone was an insufficient nutrient source for high-yielding varieties, but the integration and application of nutrients with GM and CF gains the optimum grain yield, mostly of cereal crops [33]. Nutrient supply with SA and 60 kg/ha N by fertilizer increased the leaf chlorophyll content, LAI, and NAR, and finally led to higher dry matter production and yield of rice (2.31 to 5.01 Mg/ha) as compared to that without GM and no fertilizer.

The adaptation of INM not only increased the yield of the main rice crop but also improved and sustained soil fertility and soil productivity, and it reduced environmental degradation. In our findings, we found that INM under the farm-level application of GM, FYM, CN, and tillage practices enhanced the growth, YA, and Y of hybrid rice. The same findings were considered by [34] and it was suggested that the application of CF with GM or FYM enhanced the yield and productivity of upland rice, and higher values were observed with 100% NPK along with GM, which was at a par with 75% NPK and GM during the rainy season in Himalayan hills [10]. Organic manures in the form of vermicompost, FYM, and green manures have equal potential in comparison to chemical and integrated fertilizers for rice production.

The application of FYM at the rate of 60 Mg/ha recorded higher corn biomass yields [35]. The application of 75% NPK to a soil test crop with 5 Mg/ha FYM recorded an increase of 25% mean grain yield of maize and 12% mean grain yield of succeeding chickpea over the general RD [36]. The succeeding chickpea productivity and quality were influenced by the residual effect of OM with residual management and 100% RDN in comparison to that without fertilizer under the rice–chickpea cropping system. The incorporation of green manure increased the root density and grain yield of wheat grown after rice [37]. There was a positive residual effect of green manuring (*Sesbania aculeata*) on

succeeding wheat as well as green gram crops, which were grown after rice, and a higher grain yield of succeeding wheat and gram was obtained when grown after rice [38]. The economic effect of 100% N through manure influenced the net returns (29%), benefit–cost ratio, and net energy (22%) as compared to 100% N through CF in the rice—chickpea CS [39,40], and it was found that the residual effect of RDF with 10 Mg/ha FYM increased the wheat yield by 71% over the RDF. The combination of CR and FYM with CF resulted in higher terms of YA, Y by 47%, and net returns by 27% of succeeding mung beans after the rice harvest over the control. The results clearly show that the application of FYM appeared to be important for long-term nutrient management in different CSs [41,42]. The same results were observed in our study: the application of organic manures influenced the growth and yield of the whole CS.

## 5. Conclusions

The integrated use of GM with CF and FYM is the sustainable approach for producing a higher yield of rice and succeeding baby corn crop. From the consideration and analysis of the experimental data, it can be concluded that the incorporation of *Sesbania* or water hyacinth as a GM in the fallow period of the rice–baby corn cropping system increased the YA and yield of both crops. The application of nitrogen along with organic source FYM reduced the consumption of chemical fertilizer by 50–60% under the rice–baby corn cropping system without reducing the yield of both crops. The application of *Sesbania* as GM and $RDN_{100}$ with 50% RDF gave the highest net returns, B:C ratio, system productivity, and system profitability under the CS. Thus, it can be concluded that the application of GM by *Sesbania* or WH along with CF and FYM reduced the CF consumption and sustained the productivity and profitability of the rice–baby corn CS. Hence, the diversification/intensification of the rice–baby corn–summer green manuring cropping system should be provided with 100% INM with FYM and CF in order to sustain the cropping system and higher yield and profitability. Consequently, based on the study, *Sesbania aculeata* incorporation with $RDN_{100}$ could be a suitable nitrogen management for improving system productivity and profitability as compared with other INM treatments in the Indo-Gangetic plain region.

**Author Contributions:** Methodology: V.K.; data curation and software: N.R.; writing—original draft preparation: V.K.; writing—review and editing: M.S.; supervision: M.K.S. All authors have read and agreed to the published version of the manuscript.

**Funding:** This work received no external funding.

**Acknowledgments:** We convey special thanks to S.P. Singh (Department of Agronomy, I. Ag. Sci., BHU, Varanasi), Surendra Singh Siwach, Dean of Agricultural Sciences, GLA University, Mathura. The first author is grateful to BHU, Varanasi for fellowship during her doctoral programme.

**Conflicts of Interest:** The authors declare no conflict of interest.

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
