# Peer review of "Rice (Oryza sativa L.)–Baby Corn (Zea mays L.) Cropping System Response to Different Summer Green Manuring and Nutrient Management"

_agronomy, doi:10.3390/agronomy12092105_

Round 1

Reviewer 1 Report

The manuscript with the title "Rice (Oryza sativa L.) - baby corn (Zea mays L.) cropping system response to different summer green manuring and nutrient management" presents a topical issue due to the increase in the global consumption of rice and baby corn.

A two-year field study was performed to assess the effect of GM and nitrogen management (NM) on yield attributes (YA), the yield of rice, and their residual (R) effect with the half-recommended dose of fertilizers on succeeding baby horn crop. Results illustrated the complementary effects of summer green manuring in conjunction with the INM in maximizing the yield attributes, yields of rice, its residual effect on succeeding baby corn, and the economics of the system.

The abstract is comprehensive. I recommend the explanation of abbreviations for the first use in the text, for example GM.

For keywords, I recommend using different words than the title for better visibility

The introduction is brief. It presents relevant information.

The Materials and Methods section is properly detailed with the exception of 2.7, where please elaborate

The results are detailed, statistically interpreted and easy to follow.

The discussion section is concise, it could be improved

The conclusion is short and comprehensive. I would suggest a broader, more detailed conclusion of the entire study.

Author Response

Point 1: Line 24. Write what is INM before abbreviation

Response 1: It has been added in line 22.

Point 2: Lines 46 to 48: use this information in discussion NOT inthe introduction

Response 2: The information has been added in discussion with some changes in line 283

Point 3: Line 60: delete [by [12]] and add ref. nu. At the end ofthe sentence

Response 3: It has been done in line 63

Point 4: what about the temperature in winter of 2019?

Response 4: The temperature data of winter 2019 has been added in line 125. It was skipped by mistake.

Point 5: In figure 1: Write the month name correctly

Response 5: The month names were not visible because the figure size, it is visible when you increase the figure size and that has been done.

Point 6: Line 145: Results instead of Result

Response 6: It has been done in line

Point 7: In Table 2, 3, 4 and 5: write the probability as <0.001 NOT 0.001 or 0.00. Add footnote to indicate the letters ofsignificance are among values at rows or columns

Response 7: The probability has been changed to <0.001 in Table 2, 3, 4 and 5. The comment regarding the footnote is unclear.

Point 8: Figure 2 and 3: they are with low resolution. They shouldbe with high resolution to be clear for the readers

Response 8: It has been changed.

Point 9: Line 263: explain why Chinese milk has this effect and its correlation with your results.

Response 9: It has been changed accordingly in line 262

Point 10: References: they should be revised because most ofthem not abbreviated correctly as nu. 12 [Archi. of Agro.and S. Sci., ] OR not abbreviated as nu. 11.

Response 10: It has been changed accordingly.

Reviewer 2 Report

The Manuscript [agronomy-1877572] entitled (Rice (Oryza sativa L.) - baby corn (Zea mays L.) cropping system response to different summer green manuring and nutrient management) provide important knowledge about the complementary effects of summer green manuring in maximizing the yield attributes, yields of rice, its residual effect on succeeding baby corn, and the economics of the system. All experiments are well designed and explained. The manuscript has good data and results those are introduced and written very well. Here, some comments for the authors that are considered as minor revision

1-     Line 24. Write what is INM before abbreviation

2-     Lines 46 to 48: use this information in discussion NOT in the introduction

3-     Line 60:  delete [by [12]] and add ref. nu. At the end of the sentence

4-     Line 130: what about the temperature in winter of 2019?

5-     In figure 1: Write the month name correctly

6-     Line 145: Results instead of Result

7-     In Table 2, 3, 4 and 5: write the probability as <0.001 NOT 0.001 or 0.00. Add footnote to indicate the letters of significance are among values at rows or columns

8-     Figure 2 and 3: they are with low resolution. They should be with high resolution to be clear for the readers

9-     Line 263: explain why Chinese milk has this effect and its correlation with your results.

10- References: they should be revised because most of them not abbreviated correctly as nu. 12 [Archi. of Agro. and S. Sci., ] OR not abbreviated as nu. 11.

Author Response

Point 1: The abstract is comprehensive. I recommend the explanation ofabbreviations for the first use in the text, for example GM.

Response 1: It has been changed accordingly

Point 2: For keywords, I recommend using different words than the title for better visibility

Response 2: some more keywords other than the title has been added in line 29.

Point 3: The introduction is brief. It presents relevant information.

Response 3: Minor mistakes has been rectified.

Point 4: The Materials and Methods section is properly detailed with the exception of 2.7, where please elaborate

Response 4: There isn’t any 2.7 section in materials and methods, it is elaborated up to 2.4.

Point 5: The results are detailed, statistically interpreted and easy to follow.

Response 5: some minor grammar hs been done in this section.

Point 6: The discussion section is concise, it could be improved

Response 6: Some improvements has been done in this section for a better understanding

Point 7: The conclusion is short and comprehensive. I would suggest a broader, more detailed conclusion of the entire study.

Response 7: An attempt has been made to make the conclusion more detailed and easier to understand in line 335.